# CoTFormer: More Tokens With Attention Make Up For Less Depth

**Amirkeivan Mohtashami**[*]
EPFL

**Matteo Pagliardini**[*]
EPFL

**Martin Jaggi**
EPFL

## Abstract

The race to continually develop ever larger and deeper foundational models is underway. However, techniques like the Chain-of-Thought (CoT) method continue to play a pivotal role in achieving optimal downstream performance. In this work, we establish an approximate parallel between using chain-of-thought and employing a deeper transformer. Building on this insight, we introduce CoTFormer, a transformer variant that employs an implicit CoT-like mechanism to achieve capacity comparable to a deeper model. Our empirical findings demonstrate the effectiveness of CoTFormers, as they significantly outperform larger standard transformers.

## 1 Introduction

Large foundational models have demonstrated remarkable performance across various tasks, predominantly employing the Transformer architecture (Vaswani et al., 2017). The ability to tackle new tasks in zero-shot or few-shot (Brown et al., 2020) settings has been attributed to emergent properties that become increasingly prominent with model size (Wei et al., 2022a). This observation has sparked a race to build progressively larger models (Touvron et al., 2023a,b; Brown et al., 2020; OpenAI, 2023).

However, despite the evident improvement in performance with size, certain challenges persist even in very large and deep models. One example is their proficiency in mathematical tasks (Cobbe et al., 2021). In response to these challenges, an alternative approach called Chain-of-Thought (CoT) (Wei et al., 2022b) has been proposed, requiring models to think step by step and articulate their thought processes, showing remarkable success (Kojima et al., 2022).

In this work, we draw attention to the intrinsic connection between constructing deeper Transformers and employing CoT. Specifically, we demonstrate that applying CoT with $n$ thought tokens approximates an $n$-times deeper Transformer with weight tying implemented on every nth layer (see Figure 1). Although similar weight tying schemes have been explored in the past (Dehghani et al., 2019), there is a distinction between CoT and conventional weight tying that we show to be critical. This distinction lies in the attention mechanism's capability to access previous CoT tokens.

Based on these observations, we propose CoTFormer, a Transformer that implicitly applies a similar mechanism to CoT. We empirically show that using CoTFormer allows us to obtain much better performance than standard models with more depth.

## 2 Method

Using Chain of Thought tokens improves model's performance on many tasks (Wei et al., 2022a). The process results in the generation of thought tokens in addition to the normal tokens. These thought tokens are generated in the same manner as the normal tokens using auto-regressive decoding. Notably, the whole process of generating thought tokens and finally generating the next normal token is similar

---

[*]Equal contribution. order is alphabetical.

Workshop on Advancing Neural Network Training at 37th Conference on Neural Information Processing Systems (WANT@NeurIPS 2023).

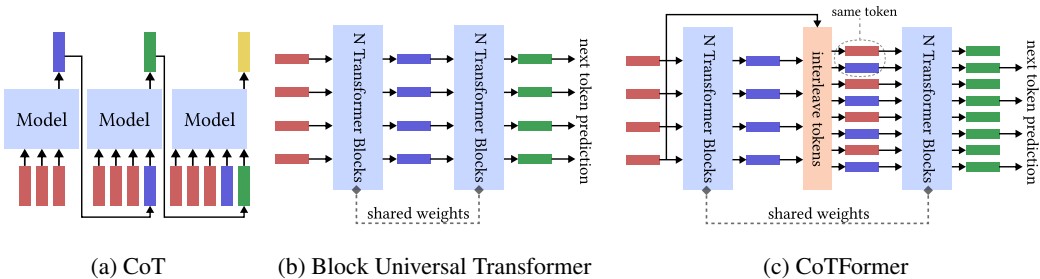

|  (a) CoT | (b) Block Universal Transformer | (c) CoTFormer |

Figure 1: **Block universal transformer vs. CoTFormer vs. Chain-of-Thought (CoT) reasoning.** In **(a)** we represent the chain-of-thought mechanism in which a model is iteratively generating reasoning tokens to help solve downstream applications. Based on existing input red tokens, a next token (blue) is generated and added to the sequence, re-iterating this process yields the green and yellow tokens. Here we emphasize how (i) the last red tokens is "pushed" several times through the model—the yellow token being the red token after three successive applications of the model—and (ii), new (e.g. blue) tokens can attend to previous (e.g. red) tokens, this observation is the basis of CoTFormer. In **(b)** we represent the block-universal transformer which recursively applies the same $N$ transformer blocks to the input tokens. This approach is to be contrasted with the CoTFormer architecture **(c)** which interleaves old and new representations in between each block. In the figure this process is done two times ($n_{\text{repeat}} = 2$), but could be repeated many more times. As in CoT, and unlike block universal transformers, later (e.g. blue) tokens can attend to earlier (e.g. red) tokens.

to decoding by recursively applying the same model multiple times (similar to a Recurrent Neural Network, RNN) (see Figure 1a). However, there is one critical distinction: Using the attention mechanism (Vaswani et al., 2017) the generated thought tokens can directly attend to previous thought tokens.

We note that if we ignore this distinction, the process of generating $n_{\text{repeat}}$ thought tokens before generating the next normal token closely resembles using an $n_{\text{repeat}}$ times deeper model. Indeed, this is the motivation behind Universal Transformers (Dehghani et al., 2019) where a single layer is recursively applied multiple times to the input to emulate an arbitrary deep model (see Figure 1b)[2].

Based on the above observation, it is possible to consider chain of thought as a poor man's version of deeper models. In this work, we argue that this is **not** the case, specifically due to the distinction highlighted above that the generated thought tokens can attend to previous tokens. To that end we propose and evaluate the following method which we name CoTFormer.

Given an input sequence $\boldsymbol{x}^{(0)} = [x_1^{(0)}, \ldots, x_{n_{\text{seq}}}^{(0)}]$ of $n_{\text{seq}}$ tokens, we pass the sequence through the model $f$ multiple times. Each pass will produce an output $\boldsymbol{y}^{(k)}$ of $n_{\text{seq}}$ new tokens that we insert back into the sequence by interleaving them with the current input $\boldsymbol{x}^{(k)}$ to build $\boldsymbol{x}^{(k+1)}$. For instance, we interleave $\boldsymbol{y}^{(0)} = f(\boldsymbol{x}^{(0)})$ to obtain $\boldsymbol{x}^{(1)} = [x_1^{(0)}, y_1^{(0)}, x_2^{(0)}, y_2^{(0)}, \ldots, x_{n_{\text{seq}}}^{(0)}, y_{n_{\text{seq}}}^{(0)}]$. We observe that each initial token has now two representative tokens. During the generation we call the last intermediate token generated for each normal token, the *active token* for that normal token. Initially the active token for each normal token is the token itself. One could imagine that the sequence length would double after each interleaving, as the input size seems to double. However, we only generate outputs for the $n_{\text{seq}}$ active tokens, which enable the sequence to grow linearly instead of exponentially as $n_{\text{repeat}}$ increases. To continue our example, we would generate $\boldsymbol{y}^{(1)} = f(\boldsymbol{x}^{(1)})$ by forwarding the $2n_{\text{seq}}$ tokens of $\boldsymbol{x}^{(1)}$ through $f$ and keeping only the outputs corresponding to active tokens. Finally, the interleaving step would yield $\boldsymbol{x}^{(2)} = [x_1^{(0)}, y_1^{(0)}, y_1^{(1)}, \ldots, x_{n_{\text{seq}}}^{(0)}, y_{n_{\text{seq}}}^{(0)}, y_{n_{\text{seq}}}^{(1)}]$, we now have 3 token representatives for each initial token. After the final round, the output for that round's active tokens are expected to be the next normal token. As such the loss is applied only on these $n_{\text{seq}}$ tokens whereas the model can arbitrarily choose what to output as intermediary tokens.

As a result of the above process, when an intermediary token is the active token and is used to generate the next output, it can attend to previous intermediary tokens.

---

[2]For the purpose of this paper, we overlook the dynamic depth of universal transformers and always assume fixed and equal depth for all tokens.

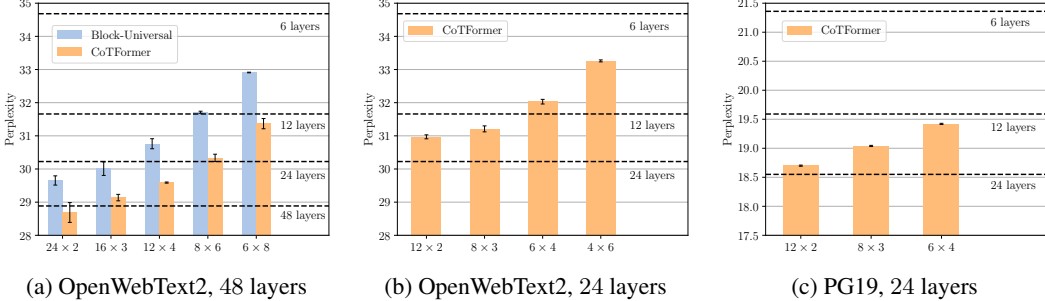

|                    |                   |              |
|--------------------|-------------------|--------------|
| (a) OpenWebText2, 48 layers | (b) OpenWebText2, 24 layers | (c) PG19, 24 layers |

Figure 2: **Comparing perplexities of different methods on OpenWebText2 and PG19.** We compare the validation perplexity obtained from training standard transformers of depth of 6, 12, 24 and 48 layers (dashed black horizontal lines), with those obtained from CoTFormer and 48 layer block-universal transformers. On the x-axis we mark $N \times n_{repeat}$ with $N$ being the number of layers and $n_{repeat}$ being the number of passes through the model (see Fig. 1). We enforce $N \times n_{repeat} = 48$ or 24, and compare with standard transformers of that same depth. In **(a)**, the CoTFormer $24 \times 2$ outperforms the standard 48-layer transformer with only half the number of parameters. Moreover, the CoTFormer consistently outperforms the block-universal transformer for a given $N$ and $n_{repeat}$. This demonstrates the positive effect of allowing attention to intermediate tokens in the CoTFormer. In **(b)** and **(c)** we match those findings when considering $N \times n_{repeat} = 24$ for both OpenWebtxt2 and PG19.

## 3 Experiments

We measure the performance of CoTFormer on two language modeling datasets: 1) PG19 (Rae et al., 2019) which is a collection of books before 1919 from Project Gutenberg. and 2) OpenWebText2 (Gao et al., 2020); a dataset of reddit submissions between 2005 and 2020. We further compare it to Block Universal Transformer to signify the importance of attention to intermediary tokens.

We train the models for 20k steps using AdamW (Loshchilov & Hutter, 2019) optimizer and apply linear warmup of the learning rate for the first 400 steps. For OpenWebText2, we use a transformer with 12 heads, embedding dimension per head equal to 64, sequence length 256, and maximum learning rate 0.001. For PG19, we use a transformer with 8 heads, embedding dimension per head equal to 128, sequence length 512, and maximum learning rate 0.001. We average our results over three seeds.

We report the final results in Table 1. The results clearly show that it is possible to obtain much superior results with a shallower model by allowing it to generate intermediary tokens using CoTFormer. Noticably, the results show a CoTFormer can outperform a standard transformer with the same depth as the CoTFormer's resembled depth. For example, a CoTFormer with 24 layers and $n_{repeat} = 2$ outperforms a standard 48 layer Transformer.

Furthermore, the bar plot of results in Figure 2 clearly shows the importance of allowing attention to intermediary tokens in CoTFormer by contrasting the significantly improved results of a CoTFormer with those of a Block Universal Transformer with the same number of repeats, both differing only by the attention to intermediary tokens.

## 4 Limitations and Future Work

While CoTFormer can obtain similar performance with smaller number of layers, the total computation cost can be higher than a deeper model due to the quadratic complexity of the attention. Still, CoTFormer significantly reduces the model size which can be useful where memory is the bottleneck and the model weights dominate memory usage. Furthermore, in this work, we focus on the setting where the number of times we pass a token through the model, i.e. $n_{repeat}$, is the same. However, similar to depth adaptive transformers Dehghani et al. (2019); Elbayad et al. (2019), it is possible to have different number of passes depending on the token which can further improve the performance. We are working on making this architecture more efficient and leave it as a future direction for research.

## 5 Related Works

**Parameter Sharing in Transformers** Dehghani et al. (2019) proposed Universal Transformers which apply a single universal layer multiple times in order to obtain the next token, simulating an

Table 1: **Performance of CoTFormer, Block Universal Transformer and Standard Transformers on PG19 and OpenWebText2**. The resembled depth of a model is assumed to be its depth times its $n_{\text{repeat}}$. The mean of 3 runs is reported with the standard error in paranthesis. It can be seen that CoTFormer highly outperforms a Block Universal Transformer with the same resembled depth and has a performance close to variants of the standard architectures that are much are deeper than its real depth. Remarkably, CoTFormer with 24 layer and 2 repeats outperforms a standard 48 layer model.

| Model | Layers | $n_{\text{repeat}}$ | OpenWebText2 | PG19 |
|---|---|---|---|---|
| Standard | 6 | - | 34.68 (0.07) | 21.36 (0.03) |
| | 12 | - | 31.66 (0.11) | 19.59 (0.02) |
| | 24 | - | 30.22 (0.09) | 18.55 (0.02) |
| | 48 | - | 28.89 (0.11) | - |
| CoTFormer (resembling 24 layers) | 4 | 6 | 33.26 (0.03) | 20.17 (0.01) |
| | 6 | 4 | 32.03 (0.07) | 19.42 (0.01) |
| | 8 | 3 | 31.21 (0.09) | 19.04 (0.01) |
| | 12 | 2 | 30.97 (0.06) | 18.70 (0.01) |
| Block Universal Transformer (resembling 48 layers) | 6 | 8 | 32.91 (0.01) | - |
| | 8 | 6 | 31.71 (0.02) | - |
| | 12 | 4 | 30.76 (0.09) | - |
| | 16 | 3 | 30.01 (0.12) | - |
| | 24 | 2 | 29.65 (0.08) | - |
| CoTFormer (resembling 48 layers) | 6 | 8 | 31.37 (0.09) | - |
| | 8 | 6 | 30.34 (0.06) | - |
| | 12 | 4 | 29.59 (0.01) | - |
| | 16 | 3 | 29.14 (0.06) | - |
| | 24 | 2 | 28.69 (0.17) | - |

artificial depth. This architecture has been extended to cases where the base model has more layers. The Block Universal Transformer architecture we explored in this work as a baseline is an example of such architecture while other weight tying arrangements are possible and are explored in Takase & Kiyono (2021). In these methods, the artificial depth is determined separately for each token. The varying depth between tokens leads introduces the problem of missing information from tokens that terminated in an early layer when using the attention mechanism on deeper layers. Various approaches have been proposed to address this issue such as copying the latest layer representation forward Liu et al. (2021). In contrast, no such problem exists in CoTFormer since going deeper is equivalent to producing a new implicit thinking token. We leave using CoTFormer in depth-adaptive architectures as a future direction.

**Chain of Thoughts** Instructing the model to write thoughts is widely used to improve the model's answers on a variety of tasks Wei et al. (2022a); Kojima et al. (2022). The efficacy of this method has been shown on different scales and the need for it does not go away even in larger transformers (Cobbe et al., 2021). In this work, we draw a parallel between generating thought tokens using auto-regressive decoding and increasing depth and show this can be used to train shallower models with higher performance.

## 6    Conclusion

In this work, we introduce CoTFormer, an architecture enabling the generation of intermediary tokens during next-token prediction, akin to manually prompting the model using the chain-of-thought method. We establish a parallel between CoT's application and the increase of model depth, offering partial insight into the success of both CoT and CoTFormer. More importantly, we emphasize the distinction between iteratively applying the model and performing CoT, highlighting the model's capacity to attend to prior intermediary tokens within CoT. We show the pivotal role of this capability in CoT-Former's superior performance by showing it significantly outperforms Block Universal Transformers.

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
