# OpenReview forum: "CoTFormer: More Tokens With Attention Make Up For Less Depth"
_NeurIPS.cc/2023/Workshop/WANT — WANT@NeurIPS 2023 Oral_

### Official Review · Reviewer_4rFe · 2023-10-23
**The paper studies efficient use of so-called Chain-Of-Thought(**CoT**) mechanism, which as the original paper(https://doi.org/10.48550/arXiv.2201.11903) states: "*a series of intermediate natural language reasoning steps that lead to the final output*". This mechanism is applied to Transformer-like architectures. The proposed architecture prevails over another CoT-exploiting and original Transformer ones in terms of parameter efficiency on seq2seq problem statement tasks.**

**Confidence:** 4

**Review:**

**Strengths**:
- Goals, methods and results seems fair and clear
- Comparing more than one dataset helps to analyse the generalisation of the proposal(each corresponds to different )
- Authors provide readable and transparent visualizations, which makes the efficiency of proposed approach clear. Perplexity metric, which is standard in seems like a reasonable choice for seq2seq problem statement.
- Sufficient number of architectures presented, addressing similar approach towards increasing Transformer's efficiency(Block Universal Transformer/CoTFormer/Original Transformer model)
- Clarity of the main method, namely CoT applied to classical NLP transformer architecture
- Hardware, datasets, hyperparameter setup are reported

**Weaknesses**:
- Frameworks in use are not mentioned
- CoT aims to increase model reasoning, which makes it appealing to test against different problem statements, i.e. Problem solving/Q&A/Summarization/etc. (*eg. https://doi.org/10.48550/arXiv.2305.10601, https://doi.org/10.48550/arXiv.2309.08589*). Though its not mentioned in future plans, it makes sense to test in order to measure the subjective/quantitative influence on * reasoning efficency* of CoT

**Recommendations for improvement**:
- Table 1: missing annotation for what metric is reported, though it's quite obvious what lays underneath
- Table 1: typo in description, row 5: "are"

---

### Official Review · Reviewer_mhT8 · 2023-10-23
**CotFormer**

**Confidence:** 3

**Review:**

Summary Of The Paper:

The paper introduces an approach for a shallow model to have performance closer to deeper models by passing the initial outputs from the model back through the model again but by this time interleaving the input and previous output tokens. This process is repeated but only the final output token will be updated so that the sequence length does not grow exponentially with the number of repeats.


Strength And Weaknesses:

Strengths:
The paper introduces a novel idea of trading off depth of transformer models with sequence length by sending the intermediary outputs of the model back through as inputs.
The charts of the paper show the improvement in accuracy clearly and the paper does not seem to claim it’s as good as increasing the depth but just shows improvement over the standard transformer.

Weakness:
The paper does not show a lot of details of how the model was trained. There is no data on what the baselines were and how long they trained each of the models for. For instance does the model work as a standard transformer that is without sending the output back into the model
It’s unclear on the advantage the paper is showing over other than it is better than a standard transformer. It seems to be worse than increasing the depth proportionally. Did it take longer to train? The answer to these questions weren’t clear.


Clarity, Quality, Originality And Significance:

Clarity: The paper presents the idea clearly with nice diagrams.

Quality: The problem the paper tackles in terms of having a smaller model reach the performance of a larger model is significant though there doesn’t seem to be a lot of details around their experiments.

Originality: The approach is an interesting idea and potentially could be something that is incorporated into future models if the training cost doesn’t change significantly for adding this.

Significance: The benefits are significant in that potentially smaller models could be used to solve problems that normally only larger models can.

---

### Official Review · Reviewer_XB4b · 2023-10-24
**Weight sharing transformers inspired by chain-of-thought prompting**

**Confidence:** 4

**Review:**

Inspired by Chain of Thought prompting (giving language models step-by-step examples in their prompt, which produces step-by-step reasoning at the output), this paper proposes passing the input tokens through the model multiple times and adding each extra output token between each input tokens at each step. In other words, if every letter in the following is a hidden vector in the transformer:

```
input:  abcd...            -f-> efgh...
step 1: ae bf cg dh...     -f-> ij kl mn op...
step 2: aej bfl cgn dhp... -f-> qrs tuv wxy zAB...
output: svyB...
```

The analogy to chain-of-thought is the additional tokens being added on each step and the additional forward passes required to run a model using chain-of-thought prompting. The authors investigate whether this design pattern will improve the performance of the model against a traditional transformer and a [Block-Universal Transformer][ut], finding that it allows transformers to approach the performance of models with many more layers.

The paper is well written and the concept is presented clearly. The inspiration of chain-of-thought makes sense in context and Figure 1 does a good job explaining the concept. The paragraph from 45-59 contains the key parts of the explanation and it is easy to read in line with the text. Results are presented clearly and relevant experimental details are available.

I think the idea that we can increase the capacity of the model by finding ways to add more tokens in the context window is an area of active interest in the community. I think the intuition is that it gives the model more activations per prediction it needs to make, and that this capacity is useful. I think there are other recent papers but I have not been able to find them while reviewing this paper.

One major issue with this method is the linear increase in the sequence length with each iteration, this increases the FLOPs required quite significantly because of the quadratic cost of the attention mechanism in the sequence length in a traditional transformer.
For the models presented in Table 1, using the [Chinchilla FLOP approximation][chinchilla] and relative to the 6 layer traditional model, the CoTFormer models require:

```
num_layers=4, n_repeat=6: 16.483391548524075
num_layers=6, n_repeat=4: 11.064310663653174
num_layers=8, n_repeat=3: 8.56763235394836
num_layers=12, n_repeat=2: 6.212862132730635

num_layers=6, n_repeat=8: 44.94020957468666
num_layers=8, n_repeat=6: 32.96678309704815
num_layers=12, n_repeat=4: 22.128621327306348
num_layers=16, n_repeat=3: 17.13526470789672
num_layers=24, n_repeat=2: 12.42572426546127
```

Pros:

- Valuable area of research with active interest
- Architecture agnostic
- Simple
- Modulates the ratio of parameters to activations
- Paper is well written
- Experimental results demonstrate good performance
- Comparison to Block-Universal Transformer is appropriate

Cons:

- FLOP cost is not quantified and is significant
- Experimental results are not comparable to large language model benchmarks

I think it's a promising method, presented well and a good candidate for discussion at a workshop.

[ut]: https://openreview.net/forum?id=HyzdRiR9Y7
[chinchilla]: https://arxiv.org/abs/2203.15556

---

### Meta-Review · Area_Chair_Wh4s · 2023-10-27

**Recommendation:** Accept (Oral)
**Confidence:** 4

**Metareview:**

**Strengths:**
* Well-written paper, with core concepts being presented clearly. Figures and visualizations were particularly praised.
* Reviewers agree that the paper targets a relevant and interesting problem.
* The proposed approach is novel, simple and straightforward to implement.
* Good evaluation with multiple architecture variants and datasets; fairly strong experimental results.

**Weaknesses:**
* Increase in sequence length due to the proposed method results in a corresponding FLOP increase. This has not been quantified and analyzed in the paper.
* Lack of training details.

The general sentiment on this paper appears to be largely positive. I recommend acceptance (poster/oral).

---

### Decision · Program_Chairs · 2023-10-28

**Decision:**

Accept (Oral)

**Comment:**

We thank the authors for their time and contribution to WANT and we are pleased to share that after the reviewing process the paper has been accepted. Congratulations! We encourage the authors to consider reviewers' feedback for the improvement of the camera-ready version. We hope to see you in person at the workshop and brainstorm on efficient training research together!